# Real-Time Correction of a Laser Beam Wavefront Distorted by an Artificial Turbulent Heated Airflow

**Alexey Rukosuev \*, Alexander Nikitin, Vladimir Toporovsky** **, Julia Sheldakova and Alexis Kudryashov**

Institute of Geosphere Dynamics (IDG RAS), 119334 Moscow, Russia; nikitin@activeoptics.ru (A.N.); topor@activeoptics.ru (V.T.); sheldakova@nightn.ru (J.S.); kud@activeoptics.ru (A.K.)
\* Correspondence: alru@nightn.ru

**Abstract:** This paper presents a FPGA-based closed-loop adaptive optical system with a bimorph deformable mirror for correction of the phase perturbation caused by artificial turbulence. The system's operating frequency of about 2000 Hz is, in many cases, sufficient to provide the real-time mode. The results of the correction of the wavefront of laser radiation distorted by the airflow formed in the laboratory conditions with the help of a fan heater are presented. For detailed consideration, the expansion of the wavefront by Zernike polynomials is used with further statistical analysis based on the discrete Fourier transform. The result of the work is an estimation of the correction efficiency of the wavefront distorted by the turbulent phase fluctuations. The ability of the bimorph adaptive mirror to correct for certain aberrations is also determined. As a result, it was concluded that the adaptive bimorph mirrors, together with a fast adaptive optical system based on FPGA, can be used to compensate wavefront distortions caused by atmospheric turbulence in the real-time mode.

**Keywords:** adaptive optics; bimorph mirror; fast adaptive optical system; turbulence; wavefront sensor; Zernike polynomials; spectral analysis

## 1. Introduction

Laser radiation, propagating in the Earth's atmosphere and beyond, allows us to solve the following tasks:

- Crypto-protected information transmission [1];
- Organization of the optical communication channels in free space [2];
- Recharging batteries of drones [3] and low-orbit satellites [4];
- Destruction of space debris [5];
- Therefore, on.

It is known [6] that air layers with different temperatures lead to the formation of turbulent refractive index changes along the propagation of the radiation. Passing through such layers, the laser beam wavefront (WF) acquires additional phase incursions, which leads to the degradation of the beam as it propagates from the source to the receiver. This limits the scope of application of the laser systems operating in a real atmosphere. One of the ways to solve this problem is to use an adaptive optical system (AOS) that is capable of compensating for the phase nonuniformity of the wavefront in real time.

At the same time, the system should have sufficient performance. As shown, for example, in [6], the frequency of phase fluctuations caused by atmospheric turbulence rarely exceeds 100 Hz. To compensate for such aberrations, a discrete AOS with a correction frequency of at least 1000 Hz (frames per second) is required. It is quite difficult to provide such a high stable performance using a conventional PC, since in addition to measuring the wavefront and calculating the voltages vector, the system should transmit information to the control unit of the deformable mirror. It is much more efficient to use a FPGA for these purposes, in which a full cycle of wavefront correction will be implemented. The use of

FPGAs will allow you to achieve a frequency that will be sufficient to correct the wavefront in real time.

It should be noted that in all previously mentioned tasks, the use of an AOS is potential, and in each case, a separate study of its applicability is required. In particular, it is necessary to solve different extra problems such as obtaining a reference signal, combating scintillations, etc.

Currently, there is a huge interest in this topic in the world. For example, in the article [7], a 9 km horizontal maritime links experiment was performed through adaptive correction. In the article [8], the issues of laser radiation correction without the use of a wavefront sensor on a 2.3 km long path with subsequent conjugation with fiber are considered. The article [9] describes an attempt to use adaptive correction in relation to the global-scale optical clock network to improve the residual instability along the 113 km path.

Before starting experiments on an open-air route, in our case, it was advisable to investigate the capabilities of the system in laboratory conditions. For this purpose, a laboratory setup was assembled, where the heated airflow from the fan heater acted as a source of distortion of the wavefront. Fourier analysis [10] applied to the data coming from the Shack–Hartmann wavefront sensor (WFS) [11,12] showed [13] that the spectral power density of the turbulent airflow is close to Kolmogorov's within the bandwidth of about 60 Hz. This parameter indicates the proximity of the laboratory conditions to the real atmosphere.

The next step in the research was the use of the expansion of the wavefront by Zernike polynomials [14]. This work is a continuation of the studies of the wavefront aberrations caused by the heat flow of the air described in [15]. Now, a similar analysis was carried out, but in the case of correction of the wavefront of laser radiation using an AOS system operating at the frequency of 2000 Hz. As a result, the efficiency of the system was evaluated, as well as the ability of the used bimorph mirror to correct for the specific aberrations.

## 2. The Fast Adaptive Optical System

The AOS includes a wavefront corrector (deformable mirror), a wavefront sensor and a control system. To achieve the specified correction speed in the experiments, a system controlled by the FPGA was used.

### 2.1. Deformable Mirror

A key element of any AOS is a wavefront corrector (WFC). It determines the ability of the system to compensate for specific aberrations, and also affects the performance of the system as a whole. When choosing the WFC, the previously measured Fried parameter [16] was taken into account, which in experiments turned out to be equal to 10 mm [17]. The measurements were carried out according to the method described in [18], where the Fried parameter was determined from the mutual dispersion of the oscillations of two focal spots of the lens array of the Shack–Hartmann wavefront sensor. The focal spots were chosen to be centrally symmetric with respect to the center of the beam. To obtain a more reliable result, several pairs of points were selected and the result of calculating the Fried parameter was averaged over these calculations.

Based on the values of the Fried parameter, it was proposed to use a bimorph mirror [19,20] as a WFC. This mirror has sufficient speed. The frequency of the first resonance is 8.3 kHz; the amplitude-phase frequency response (Bode diagram) is shown in Figure 1. The phase-frequency response of the mirror becomes equal to 90 degrees at a frequency of about 8.2 kHz, while the peak of the amplitude resonance is at 8.3 kHz.

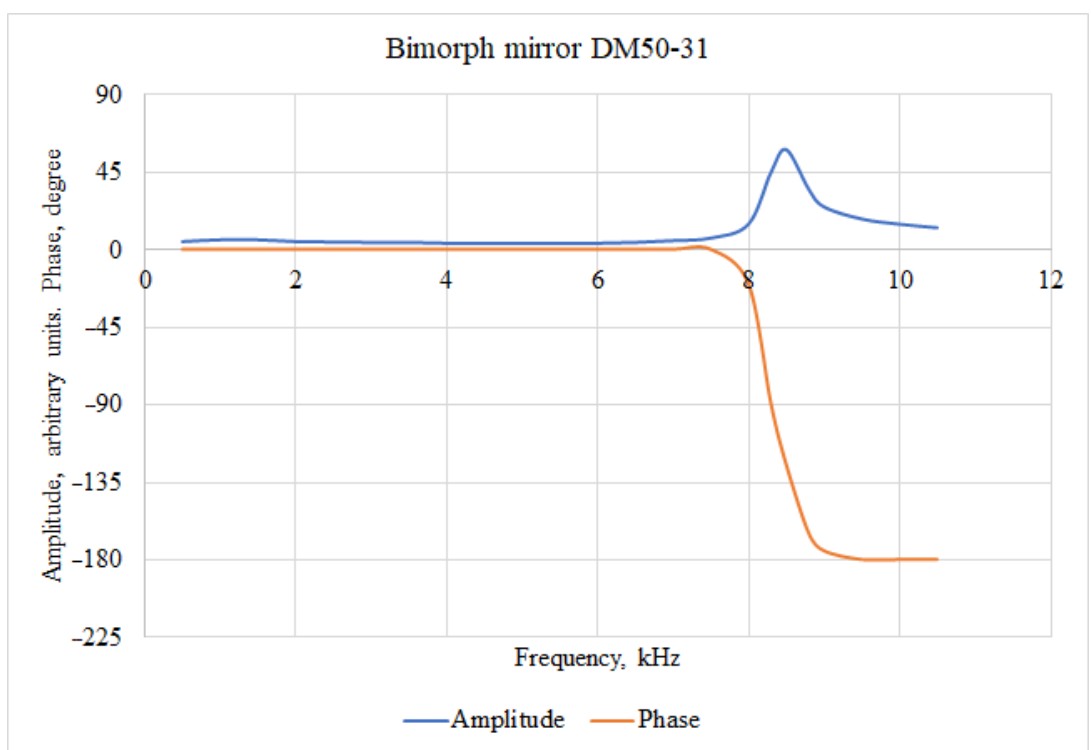

**Figure 1.** Bode diagram for bimorph deformable mirror.

The structure of the electrodes consists of three 8 mm wide rings, which should be enough to correct for the WF with the Fried parameter of 10 mm. The photo of the mirror and the structure of the electrodes are shown in Figure 2. Table 1 shows the main characteristics of the mirror.

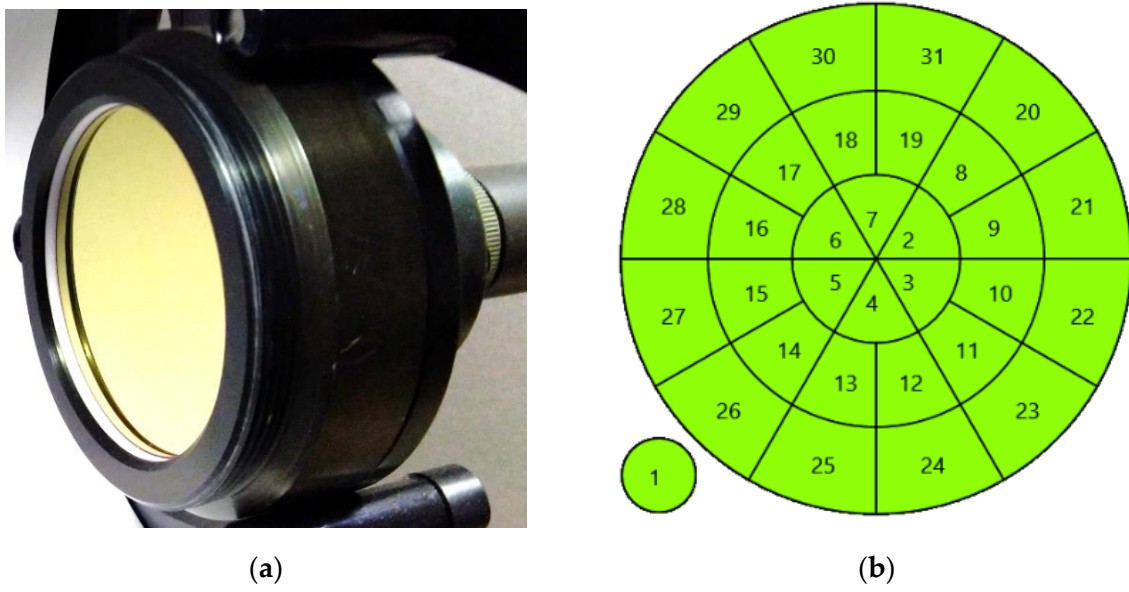

(**a**)          (**b**)

**Figure 2.** Bimorph mirror used in experiments: (**a**) Mirror photo; (**b**) electrodes structure. The numbers indicate the channels of the mirror. The first electrode, conventionally shown in the lower-left corner, has the dimensions of the mirror aperture and is used to control the defocusing of the wavefront.

**Table 1.** Main parameters of the bimorph mirror.

| Parameter | Value |
|---|---|
| Clear aperture | 50 mm |
| Electrodes number | 31 |
| Control voltage range | $-200$ V $\pm$ 300 V |
| Maximal stroke | $\pm 10$ µm |
| First resonant frequency | 8.3 kHz |
| Coating | Protected silver |
| Size | Ø 70 mm × 68 mm |
| Weight | 320 g |

*2.2. Shack–Hartmann Wavefront Sensor*

To ensure the system operating frequency of the 2000 Hz, a wavefront sensor based on a fast camera was used. The performance improvement was achieved by using a part of the image. Thus, in the experiments we used a resolution of 480 × 480 pixels, which allowed us to increase the frame rate to 4000 Hz. The main parameters of the wavefront sensor are presented in Table 2

**Table 2.** Main parameters of the wavefront sensor.

| Parameter | Value |
|---|---|
| Sensor | Alexima LUX19HS |
| Spectral bandwidth | 350–1100 nm |
| Dynamic Range (Tilt) | $\pm 50\lambda$ |
| Accuracy of measurements | $\lambda/90$ |
| Frame rate | 2500 fps @ 1920 × 1080 |
|  | ~4000 fps @ 480 × 480 |
| Interface | Fiber Optic 40 Gb/s |
| Lenslet array focal length | 12 mm |
| Number of working sub-apertures | 20 × 20 |
| Input light beam size | 4.8 × 4.8 mm |
| Resolution | 8 bit |

*2.3. Adaptive Optical System Control Loop*

To ensure the fast operation of the AOS, the control loop was made using FPGA. The system requires (1) preloading of a reference—a set of coordinates of WFS focal spots [21]—to which the coordinates of the real focal spots will be pulled up using WFC; and (2) preloading of WFC response functions. Based on the analysis of the image coming from the WFS, the FPGA calculates a vector of corrective voltages, which is then applied to the mirror electrodes. To achieve a high speed of operation in the system, a phase-conjugated algorithm was used.

Since the bimorph deformable mirror cannot reproduce tilts, virtual tilts response functions are introduced into the system to exclude them. In fact, the voltages are also calculated for tilts, but they are not applied anywhere.

In the experiments, the AOS operated with a correction frequency of 2000 Hz (frames per second) in the closed-loop mode. To obtain such a high-speed performance, FPGA Arria V GZ processed the image bytes coming from the WFS camera 'on the fly' (as they arrive from the camera), which made it possible to calculate the vector of control voltages by the end of the frame reception. Since the sensor camera operated at a frequency of 4000 Hz, it took 250 microseconds. One hundred and fifty microseconds were required to transmit the voltage vector to the adaptive mirror Control Unit, fifty microseconds to load the DAC, and the pause required for the mirror to change its shape lasted fifty microseconds. The internal structure of the FPGA is shown in Figure 3, and the corresponding time diagram of its operation is shown in Figure 4.

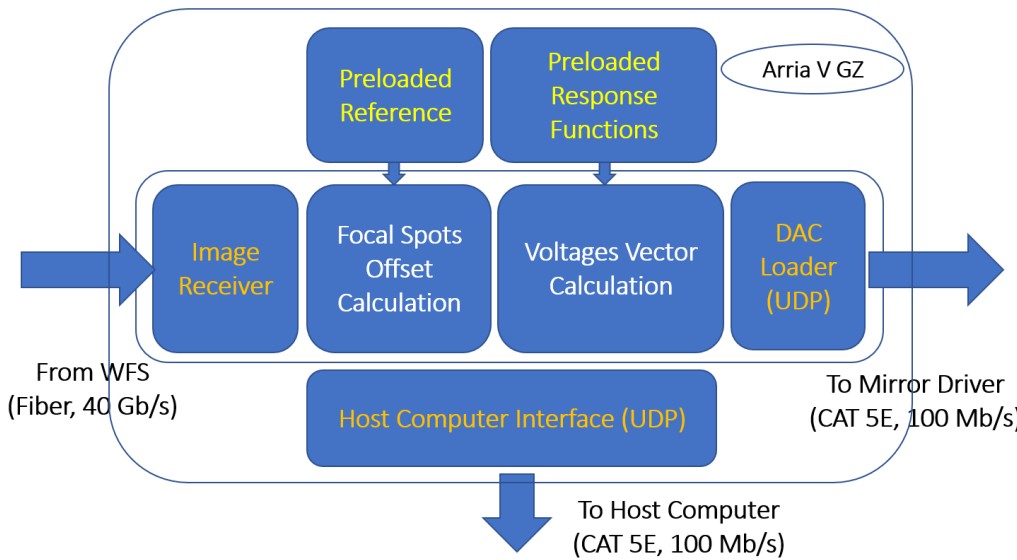

**Figure 3.** FPGA inner structure.

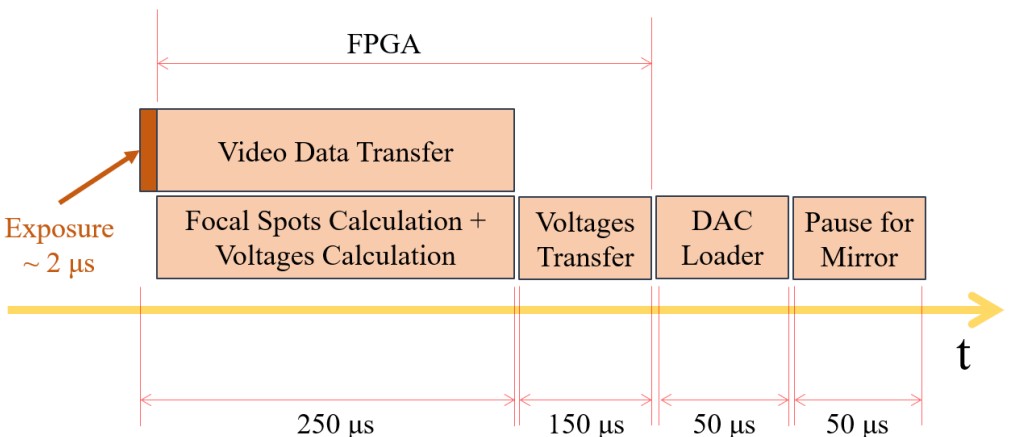

**Figure 4.** FPGA timing diagram. The total time of one closed cycle is 500 microseconds, which corresponds to an operating frequency of 2000 Hz.

## 3. Experimental Setup

To correct for the WF, a laboratory installation was used, as shown in Figure 5.

The source of the radiation is a laser diode coupled to an optical fiber. The wavelength is chosen as 650 nm to facilitate the alignment. The collimating lens forms a parallel beam with a diameter of 50 mm. A fan heater is installed on the path of propagation of the laser beam, the heated airflow of which crosses the laser beam, creating the turbulence. A laser beam with a distorted wavefront hits a WFC-bimorph mirror and is reflected from it in the direction of the WFS. A flat mirror installed between the WFC and the WFS is used to reduce the dimensions of the installation. Part of the beam in front of the WFS branches off to the far-field indicator formed by a long-focus lens and a CMOS camera with a small pixel size (3.75 microns) for a more detailed visualization of the image. The operation of the system is controlled by the FPGA, which performs all the functions necessary for the correction of the wavefront: it receives an image from the WFS camera, calculates the vector of correcting voltages and transmits this vector to the WFC amplifier unit (Mirror CU). The PC in this configuration performs only the functions of controlling the operation of the FPGA and the functions of monitoring the correction process.

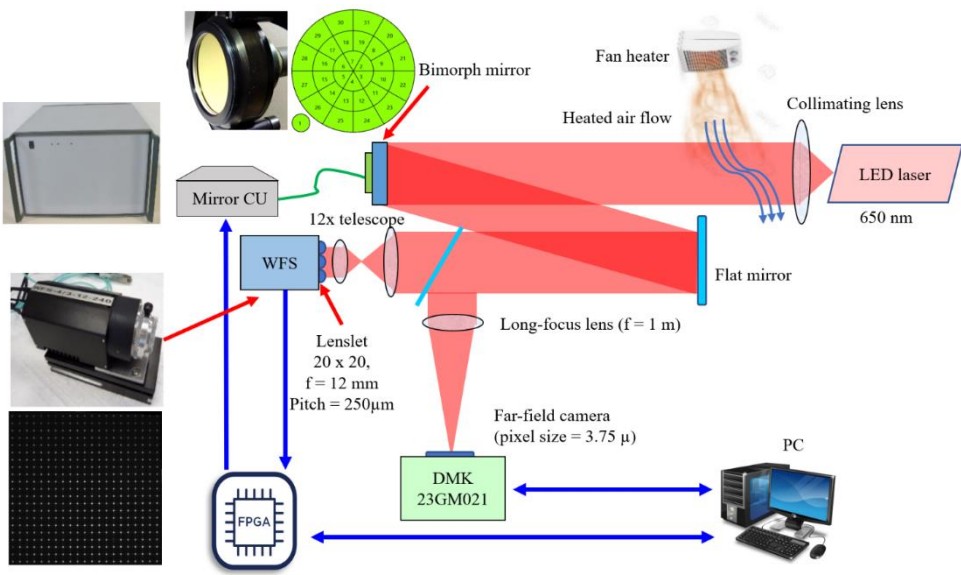

**Figure 5.** Adaptive optical system experimental setup. WFS—wavefront sensor; mirror CU—mirror control unit.

## 4. Processing the Results of the Experiment

Before starting the work, studies of distortions of the wavefront of the laser beam caused by the influence of a heated air stream were carried out. Figure 6 shows the spectral power density of the process obtained on the basis of statistical processing of a series of recorded fluctuations in the coordinates of the focal spot of the WFS lens array. The straight line $f^{(-5/3)}$ corresponds to the Kolmogorov spectrum. The sampling duration of the focal spot coordinates was 10 s (at a frequency of 2000 Hz, 20,000 samples were recorded), which provided a resolution along the frequency axis of 0.1 Hz.

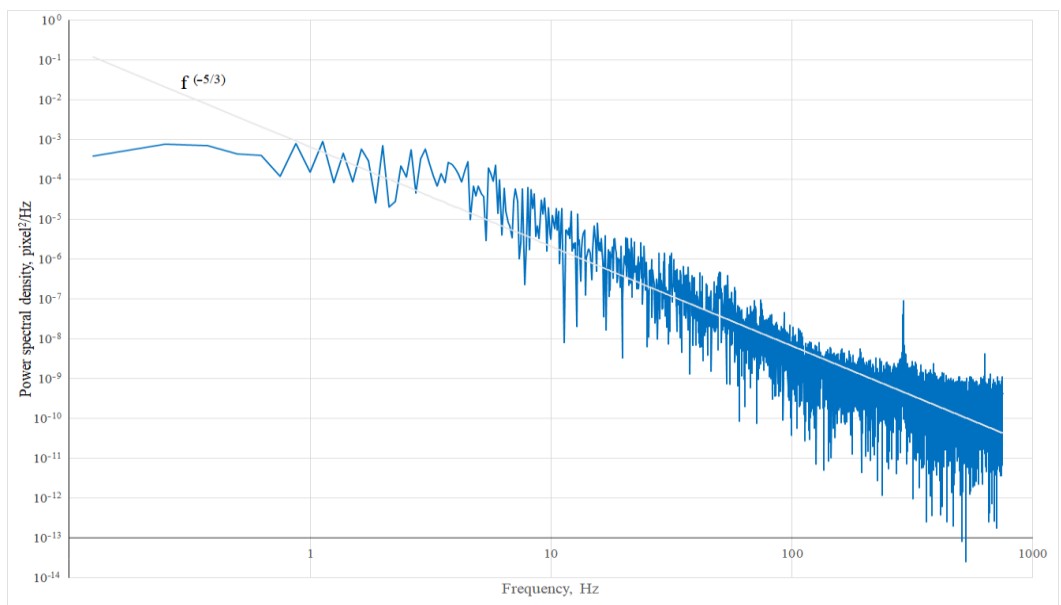

**Figure 6.** Spectral power density of the oscillation of the X coordinate of the WFS lens array focal spot.

The quality control of the correction was carried out on the far field image [22]. With the specified parameters of the laboratory setup (beam diameter of 50 mm, lens focus length of the far field zone indicator of 1 m and radiation wavelength of 650 nm), the diffraction-limited diameter of the spot in the lens focus was about 32 microns. Correction of the laser

radiation wavefront makes it possible to obtain a focal spot diameter in the far field at the level of 9 pixels, which, with a pixel size of 3.75 microns, corresponds to 34 microns. Thus, the diameter of the spot in the far field zone is close to the diffraction limit and, consequently, the correction quality is quite good. Figure 7 shows images of the focal spot in the far field. The upper pictures were obtained in the absence of correction, while the lower picture shows the intensity distribution in the presence of wavefront correction. It should be noted that in the absence of correction, the intensity of the focal spot was too low, so the upper row of images was obtained with a longer exposure (85 µs vs. 57 µs for the corrected case). The image of the far field is close to the diffraction one and practically does not change its shape during the correction procedure.

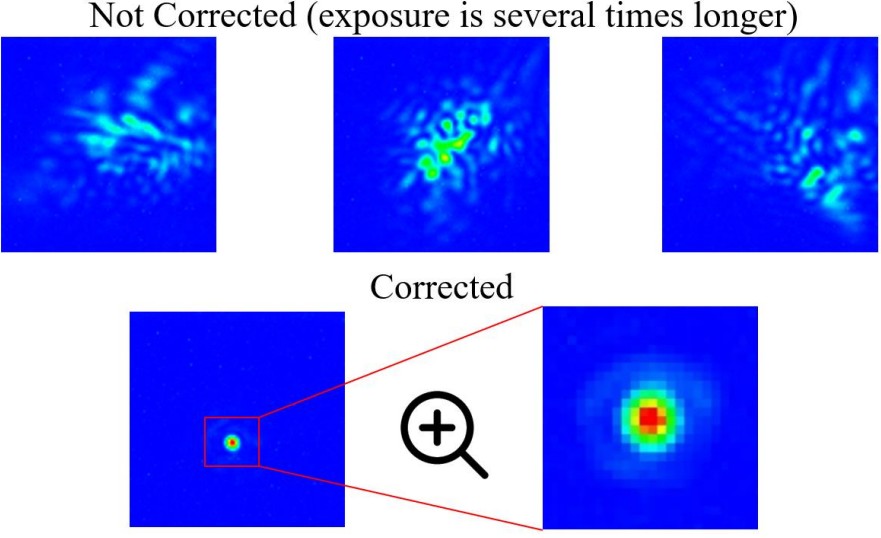

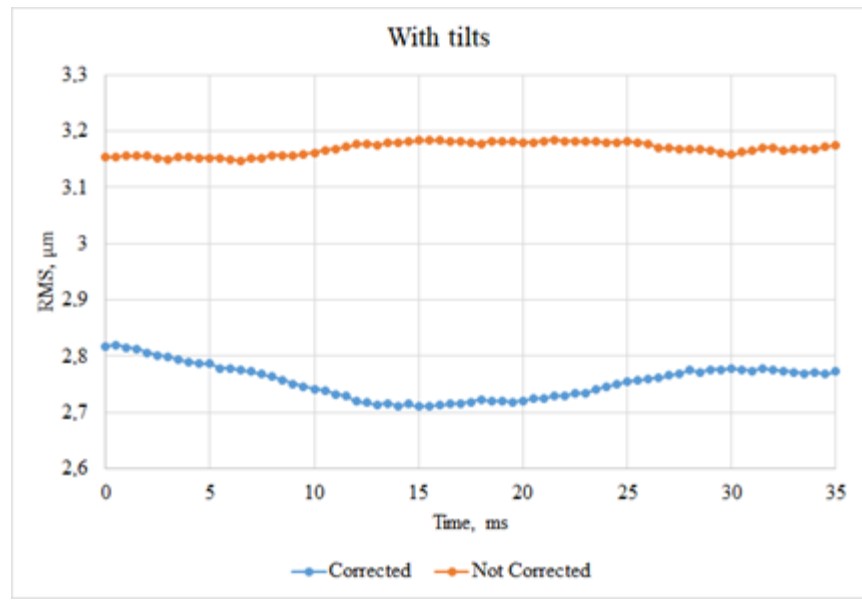

**Figure 7.** Far-field images. Top row—without correction; bottom row—during correction.

Another way to assess the quality of the correction is the residual error. Figures 8 and 9 show the change in residual RMS over time for the case of correction and the absence of correction. Figure 8 represents a complete set of aberrations and graph on Figure 9 was obtained by excluding the tilts.

**Figure 8.** Residual RMS changes over time; full set of aberrations.

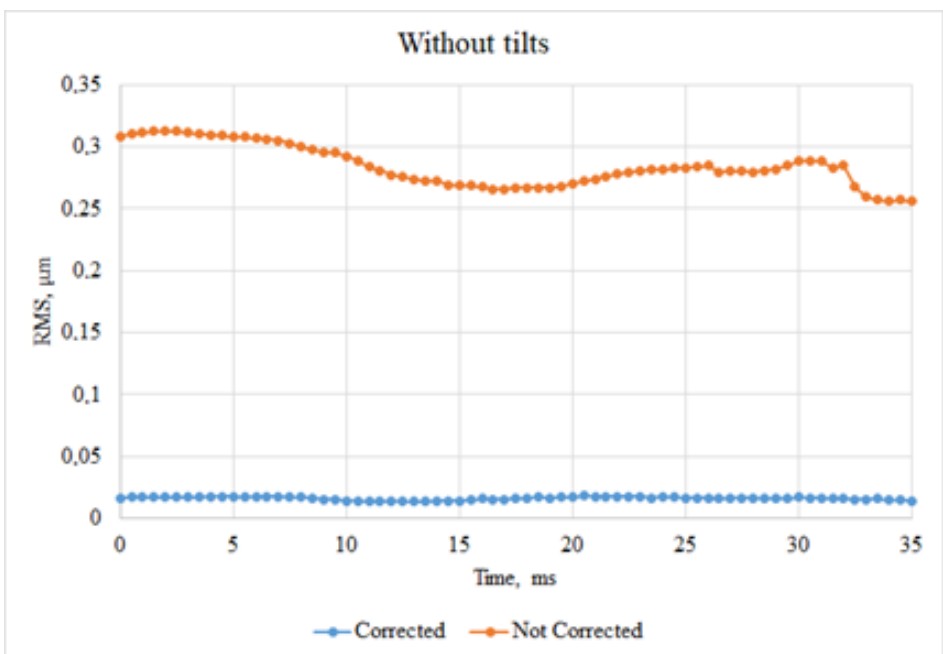

**Figure 9.** Residual RMS changes over time, without tilts.

For a more detailed consideration of the quality of correction and to obtain quantitative results, a transition was made to the spectral analysis of each of the modes of the wavefront decomposition by Zernike polynomials. The data-processing algorithm was as follows.

- A sample of the offset coordinates of the focal spots of the lens array of the WFS with a duration of 10 s was recorded. This made it possible to achieve a resolution along the frequency axis of 0.1 Hz. At a frequency of 2000 Hz, a total of 20,000 values of coordinate offsets of each focal spot were recorded
- The transition was carried out from sampling by coordinates to sampling by the coefficients of the wavefront expansion by Zernike polynomials. In this work, we used a set of 24 Zernike polynomials in Wyant indices [23] (1 and 2—tilts, 3—defocus, 4 and 5—astigmatism, 6 and 7—coma, 8—spherical, etc.).
- Using the Fourier transform, the transition from the time domain to the frequency domain was carried out for sampling each Zernike polynomial;
- The power spectral density was calculated for each mode;
- Further, by integrating the spectral power density, the spectral energy was calculated.

If we integrate the spectral power density for each Zernike polynomial, we can get a graph of the spectral energy (Figure 10). The graphs at a certain frequency value go into saturation, which indicates an insignificant contribution of higher-frequency components to the total signal energy. The frequency of transition of graphs to saturation can be taken as the bandwidth occupied by one Zernike polynomial or another.

Figure 11 shows a diagram of the dependence of the frequency band occupied by one aberration or another. The graph corresponds to one of the consequences of Taylor's hypothesis [24], according to which the rate of change of high-order aberrations is faster.

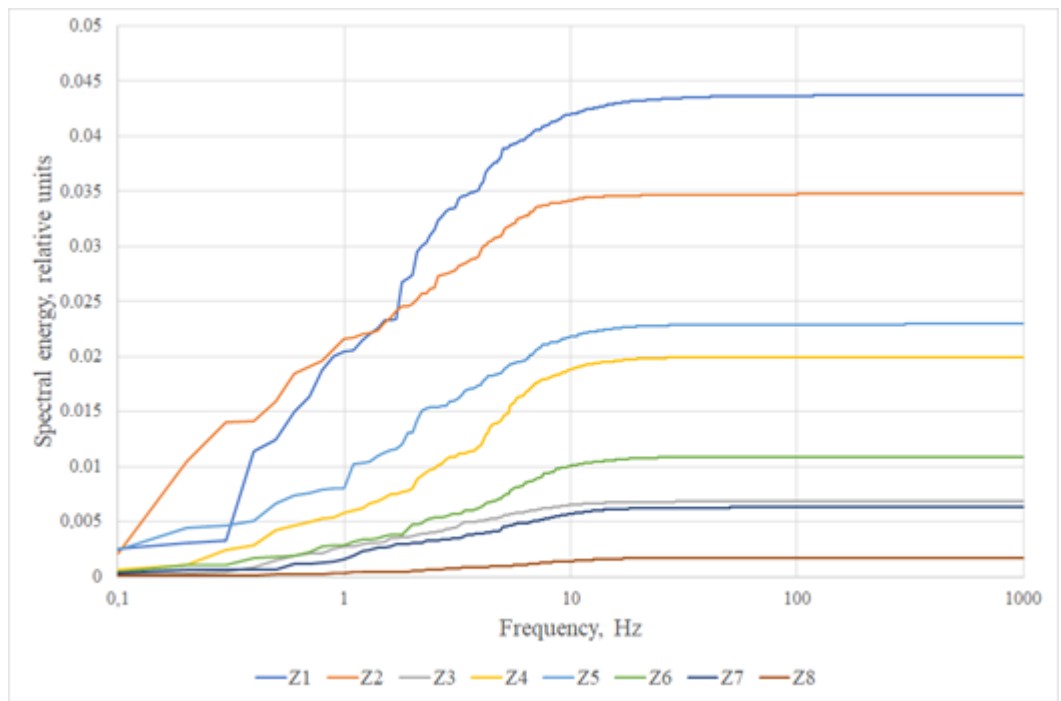

**Figure 10.** The spectral energy for first eight Zernike polynomials.

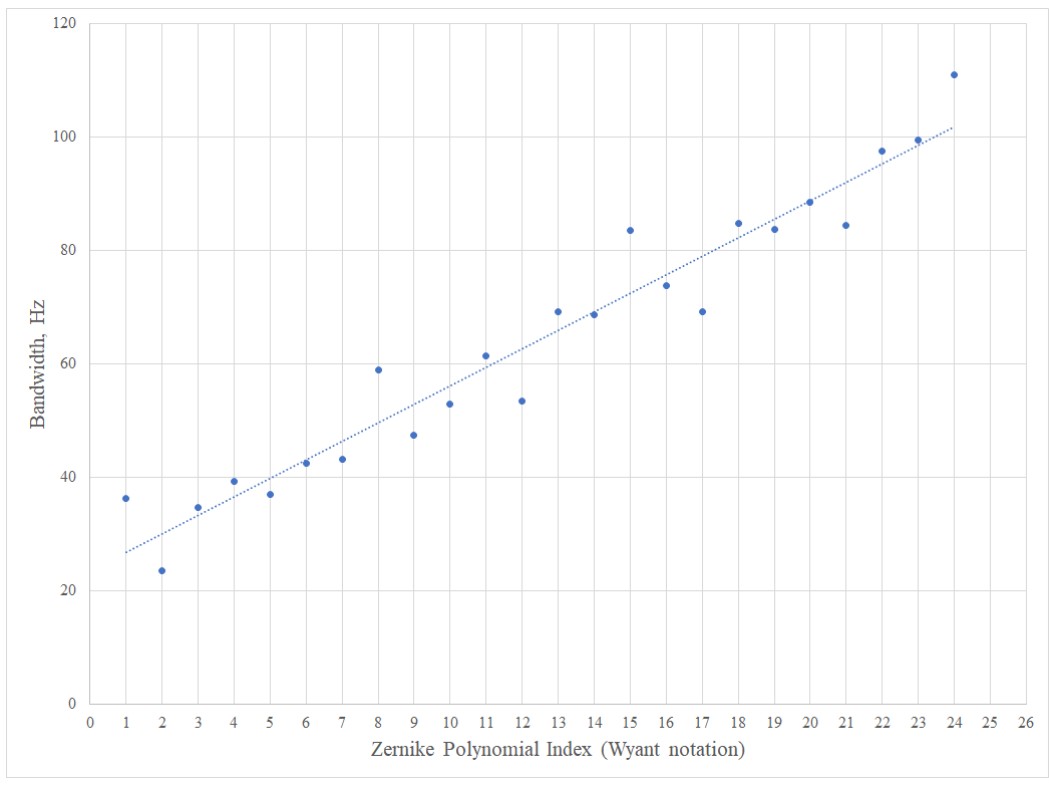

**Figure 11.** The frequency band occupied by one or another aberration.

The spectral energy saturation amplitude from Figure 10 for each Zernike polynomial is shown on Figure 12.

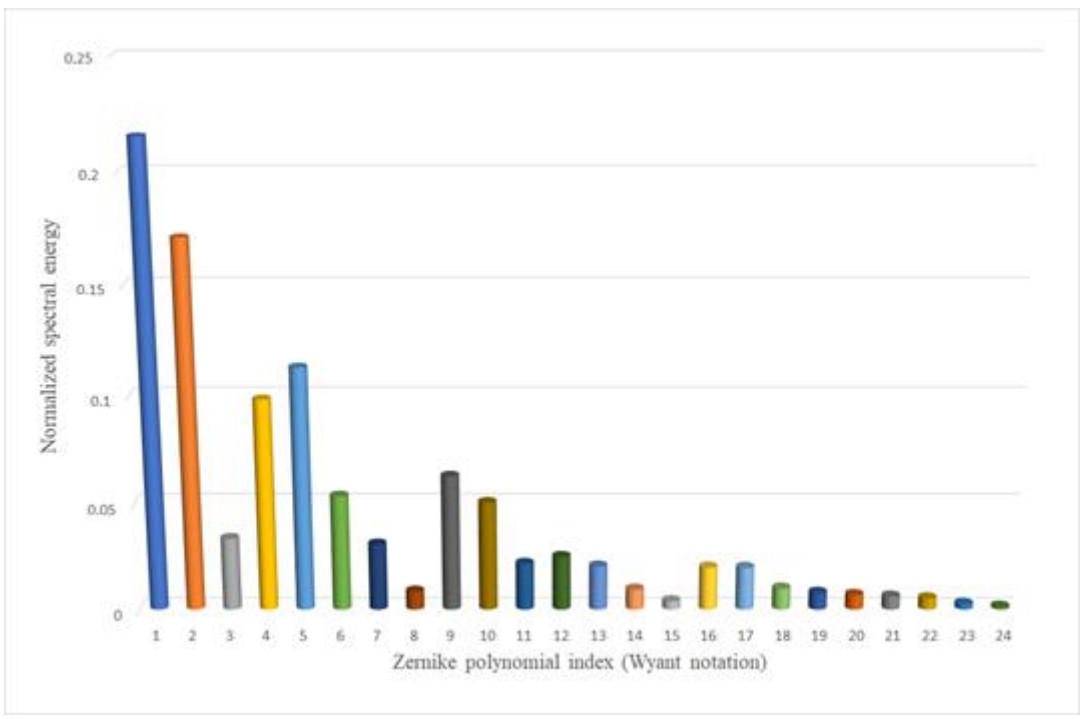

**Figure 12.** The spectral energy of wavefront aberrations before correction.

The residual aberrations are quite small, so Figure 13 represents a diagram of the uncompensated aberrations, expressed as a percentage of the entry level.

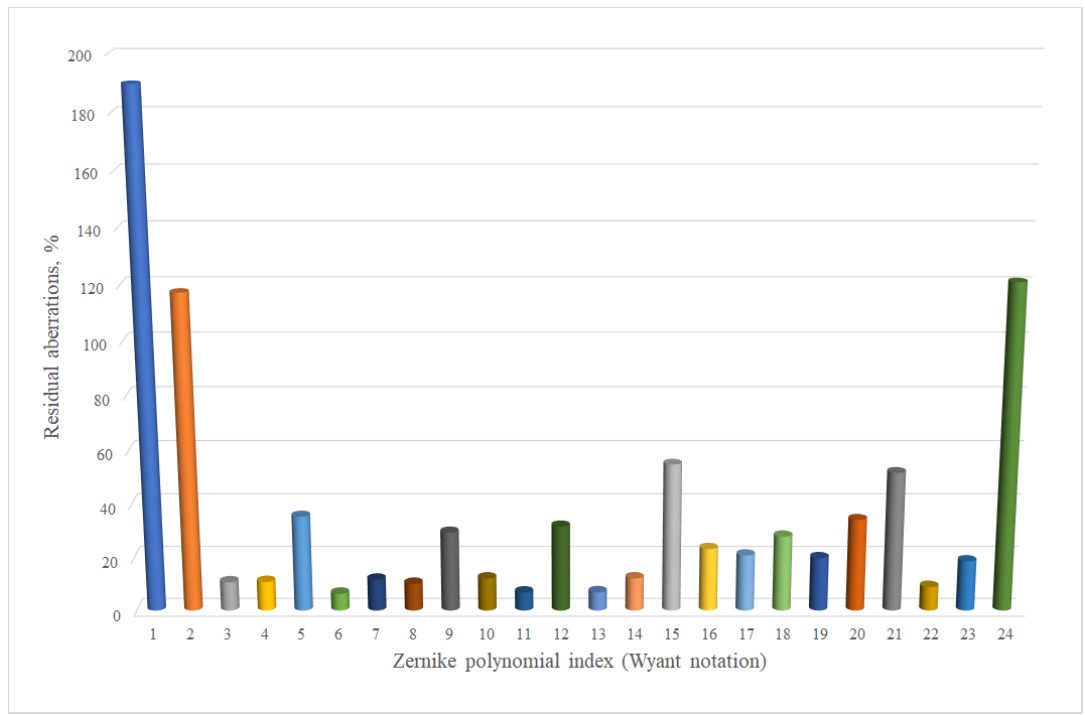

**Figure 13.** Residual aberrations after correction.

For greater clarity, Figure 14 shows the same diagram, but expressed in decibels.

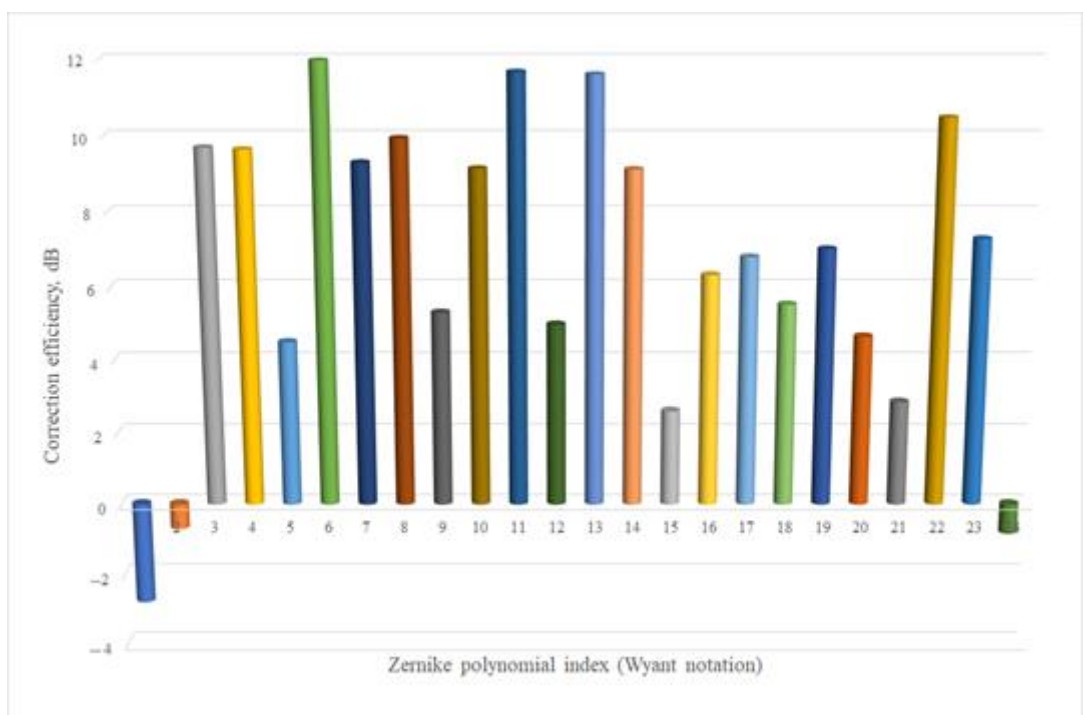

**Figure 14.** Correction efficiency for each Zernike mode.

**5. Discussion**

Based on the diagram on Figure 14, we can make the following suggestions.

1.  The combination of FPGA performance and a bimorph wavefront corrector in an adaptive optical system allows correction of artificially created turbulence in real time. The correction frequency in the experiments was chosen to be equal to 2000 Hz. Such a stable correction frequency is almost impossible to obtain using a conventional PC. The PC, unlike the FPGA, performs I/O at the driver level, thereby increasing the time of the closed correction cycle. FPGA exchanges data between external devices (wavefront sensor and corrector control unit) directly. In addition, the FPGA performs parallel processing of information, which has significant limitations in the case of using a PC.

2.  The speed of the AOS controlled by the FPGA made it possible to analyze the effectiveness of aberration correction up to the 23rd Zernike polynomial, whose bandwidth is about 100 Hz, in detail.

3.  The bimorph WFC does not have the ability to correct for the slopes of the WF. Accordingly, the correction efficiency of the first two Zernike polynomials is negative, i.e., there is an increase in the amplitude of the initial slopes. To eliminate the slopes, it is necessary to use either a separate beam position stabilization system (see, for example, [25]), or to install a mirror in a tip–tilt mount. In this case, the virtual slopes used in the experiment become real and the voltages calculated during operation are applied to the control drives of the tip–tilt mount.

4.  WFC used in the experiments has three rings of electrodes and cannot reproduce a spherical aberration of the third order (polynomial # 24), since this aberration has four extremums (max–min). To reduce the aberration # 24, a higher spatial resolution WFC is required.

5.  The aberrations from 3 to 23 are compensated well enough by this WFC. However, because the initial amplitude of the polynomial # 24 is small compared to other aberrations, the undercompensating of this aberration can be neglected.



## 6. Conclusions

We demonstrated a closed-loop adaptive optical system with the bimorph deformable mirror, Shack–Hartmann wavefront sensor and FPGA controller that can efficiently correct for the wavefront aberrations caused by the artificial turbulence. The total speed of the system operation was equal to 2000 Hz. It should be noted that the bimorph deformable mirror corrects for the wavefront aberrations caused by the flow of the heated air quite well—the residual error of the phase fluctuations was reduced by more than 10 times and an almost diffraction-limited focal spot was obtained. Certainly, when using such a type of corrector, it is necessary to additionally apply a system to stabilize the position of the beam in space.

**Author Contributions:** Conceptualization, A.K., A.R.; methodology, A.K., A.R.; software, A.R.; validation, A.K., A.N., J.S.; formal analysis, A.N., A.R., V.T.; investigation, A.R., A.N.; resources, A.K.; data curation, A.R., V.T.; writing—original draft preparation, A.R.; writing—review and editing, A.N., J.S., V.T.; visualization, A.N., A.R.; supervision, A.K.; project administration, A.K. All authors have read and agreed to the published version of the manuscript.

**Funding:** The research was carried out within the state assignment of Ministry of Science and Higher Education of the Russian Federation (theme # 122032900183-1).

**Data Availability Statement:** The data presented in this study are available on request from the corresponding author.

**Conflicts of Interest:** The authors declare no conflict of interest. The funders had no role in the design of the study; in the collection, analyses, or interpretation of data; in the writing of the manuscript, or in the decision to publish the results.

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
