# Peer review of "Real-Time Correction of a Laser Beam Wavefront Distorted by an Artificial Turbulent Heated Airflow"

_photonics, doi:10.3390/photonics9050351_

Round 1

Reviewer 1 Report

This paper describes the use of adaptive optics to correct the wavefront distortion imparted to a laser beam due to atmospheric turbulence, simulated in the lab using a fan heater. Correction of laser transmission through turbulent channels is of significant importance to communications, fundamental physics, and geodesy, among other fields.

The paper is well-written, clearly describing the experiment. However, it is not clear where the novelty lies; the system being described is not attempting anything new and is a standard adaptive optics implementation as could be found in an educational laboratory. Based on this, and without further context or justification for this particular experiment, I cannot see this paper being of interest to the wider community.

More specific comments:

  • The paper gives a very broad context for why turbulence correction is important, but does not give more specific motivation for this particular experiment. Is it part of some larger project working towards a specific application?
  • The reference concerning optical communications is a conference presentation summary from 2010, there are many more recent and substantial references available.
  • There is no mention of works by other authors, eg [1,2, 3] and references therein, or acknowledgement of the current status of the field, which has matured well beyond the level of this experiment to links across kilometres of atmosphere. [3] in particular uses AO to correct a beam over 113km of turbulent atmosphere for frequency metrology.
  • The quality of wavefront correction is described in terms of reduction of Zernike coefficients. While this shows that the system is reducing the aberrations corresponding to these modes, the metric that ultimately matters is the total remaining wavefront RMS error. Images of the focused spot showing corrected vs uncorrected would also be very helpful.
  • For the system described in this paper, the tip and tilt modes are not only uncorrected, but are significantly worsened. Since tip and tilt have the largest effect on wavefront RMS, it is probable that the corrected RMS is higher than the uncorrected RMS for this system. The authors identify that this is due to the limitations of the deformable mirror, and that a separate tip/tilt stage is needed. However, it is already established practise in adaptive optics to have a separate tip/tilt mirror or to mount the deformable mirror on some stage with higher actuation for this reason.

[1] Rui Wang, Yukun Wang, Chengbin Jin, Xianghui Yin, Shaoxin Wang, Chengliang Yang, Zhaoliang Cao, Quanquan Mu, Shijie Gao, Li Xuan, Demonstration of horizontal free-space laser communication with the effect of the bandwidth of adaptive optics system, Optics Communications, Volume 431, 2019, Pages 167-173

[2] Weyrauch, T., Vorontsov, M.A. (2004). Free-space laser communications with adaptive optics: Atmospheric compensation experiments. In: Free-Space Laser Communications. Optical and Fiber Communications Reports, vol 2. Springer, New York, NY. https://doi.org/10.1007/978-0-387-28677-8_5

[3]  Shen et al, 113 km Free-Space Time-Frequency Dissemination at the 19th Decimal Instability, arXiv:2203.11272 

Author Response

Thank you so much for your review of our paper. Your comments are really very significant for us. We tried to answer all your comments and make appropriate amendments to the text of our paper.

In fact, this work is not a part of any “major project”. All experiments were carried out within the framework of our internal interests and actual research activity in the field of adaptive optics (of course, we consider the future interest from Japanese, Singapore, European or even Chinese end-users), the purpose of which is both to study the distortions introduced by atmospheric turbulence along the horizontal path, and the possibility of correcting such distortions. Accordingly, the work does not investigate free-space communication systems, but focuses on the possibility of correcting strong Kolmogorov turbulence.

What is new in the article - the description of an experiment and the method to correct the wavefront of the laser radiation distorted by a jet of heated air with an adaptive system operating with the frequency of 2 kHz and also by using a bimorph mirror as a wavefront corrector. For the correction the phase conjugation method was considered. The simplest fan heater was used as a source of turbulence, the intensity of which was selected based on the dynamic range of the adaptive optical system. It happened so, that the spectrum of phase fluctuations introduced by this fan was … close to some extend to Kolmogorov one.

As for the new works, indeed, a lot of articles on this topic have been published recently. Accordingly, the proposed articles were added in the introduction, since they are really important and of great scientific interest. Missing of those papers was our real fault.

We also followed the suggestion and put for greater clarity the images of a focused spot to the text of the article.

As for the slopes, quite often a separate beam position stabilization system is used for this, see, for example, [1]. In addition, a similar system has been developed in our laboratory and a corresponding article is being prepared for publication. In the future, it is planned to conduct the experiments on the joint use of a beam position stabilization system and a fast adaptive optical system.

Also, the purpose of the article was to evaluate the ability of a bimorph mirror to compensate for a particular Zernike mode. To verify the correctness of the calculations, a mirror with three rings of electrodes was specially selected, which is not capable of correcting spherical aberration of the 3rd order. Experiments have confirmed this.

[1] Rudolf Saathof, Remco den Breeje, Wimar Klop, Niek Doelman, Thijs Moens, Michael Gruber, Tjeerd Russchenberg, Federico Pettazzi, Jet Human, Ramon Mata Calvo, Juraj Poliak, Ricardo Barrios, Mathias Richerzhagen, Ivan Ferrario, "Pre-correction adaptive optics performance for a 10 km laser link," Proc. SPIE 10910, Free-Space Laser Communications XXXI, 109101H (25 March 2019); https://doi.org/10.1117/12.2506849

Reviewer 2 Report

The paper describes a laboratory AO experiment conducted in a way as many have been in the mid 1990s (apart from the correction frequency, which was a couple of hundred Hz at the time).  As such, it contributes little new knowledge.

The context is laser communication, however the setup used resembles that of an astronomical AO system with the atmosphere introduced between the light source and the AO system.  In terms of a communication system, this resembles a correction at the receiving station,  whereas true communication, and more so things like space debris destruction would require correction next to, or inside the launch station.  This field has its own problems, from requiring a reference source at the receiving end to point-ahead angles when communicating with moving targets in space which results in up- and downlink travelling through different parts of the atmosphere. Another major challenge in communication is scintillation, which is strong when the largest turbulent contribution is close to the source, and which causes dramatic signal losses. Not any of these problems are mentioned in the introduction, instead a system is presented as a model which is not up to the task in any way.

So what remains is essentially a preparatory experiment to demonstrate that the combination of an SHS and a bimporph mirror can correct for aberrations introduced by turbulence - as has been done frequently around 1995.

What might be of interest is the operation of the bimorph mirror at very high frequency, however no temporal characteristics are presented with the exception of mentioning the resonant frequency. It remains a mystery however, what the characteristics really are, e.g. what percentage of a commanded stroke or shape is achieved after one loop cycle time.

Also, the control bandwidth is never determined .  I would at least have expected some plots of temporal power spectra and a transfer function for, say, 3 different Zernike terms (maybe the sphericals?). Maybe even a Bode-plot to show the effective correction bandwidth at 3 spatial frequencies?

Speaking of metrics, I am really stunned that the word "Strehl" is never mentioned anywhere in the paper, which is clearly the standard measure of image quality. Instead, the authors measure the diameter of the point spread function, which is normally useless as it becomes equal to the diffraction limited diameter for Strehl ratios above ~ 20%. That might of course be sufficient depending on the application, but it's not a good measure to judge AO performance by.  Either quote residual wavefront error, or Strehl ratio.

With all these shortcomings, it remains unclear what the goal of the paper is. Simply presenting a demonstrator for a low-order AO system is hardly news-worthy in 2022.  I suggest to either retract the paper, or to expand drastically on the parts that appear interesting, e.g. the temporal characteristics, or a more detailed description of the FPGA implementation. 

Author Response

Thank you so much for your review of our paper. Your comments are really very significant for us. Here we tried to answer all your comments and make appropriate amendments to the text of our paper.

Based on the experience of our work, we have not encountered an analysis (description) of the operation of an adaptive optical system (AOS) at frequencies of 2 kHz with a bimorph mirror as a wavefront corrector. We also do not know the results of the studies of the mode compensation of the certain aberrations. This article shows that a relatively simple corrector - a low-cost bimorph mirror is able to operate at a frequency of 2 kHz and at the same time compensate for a fairly wide range of phase fluctuations. For the demonstration, just a 32-electrode mirror with a limited spatial resolution was chosen in order to show its limitations. Thus, the article demonstrates that this type of the wavefront corrector cannot compensate for the slopes of the wavefront as well as spherical aberrations of the 3rd order and higher. It is obvious that, if necessary, the use of a wavefront corrector with a large spatial resolution will allow correcting aberrations of the higher orders.

The article focuses on the study of the quality of the wavefront correction by an adaptive optical system. The list of the possible applications includes the potential situations where such a system can be applied. In this article we are talking about distortions of the wavefront (phase), which are not transformed into intensity during propagation. Accordingly, the possible amplitude of any scintillations in this case will be small. The authors understand that adapting such a system to the real task requires a special approach, but this is beyond the scope of this publication.

As far as we know, in the mid-90s, a bimorph mirror manufactured by CILAS was supposed to be used on a Subaru telescope or may be in some other astronomic instruments. There is a paper published back in 2006 [1] describing this kind of application. In fact, we did not meet any extra papers on the use of bimorph technology to correct for atmospheric phase fluctuations.

In our AOS, the phase conjugation algorithm based on the least square method was used to calculate the control voltage vector. Ideally, the system should be able to realize the desired surface of the mirror conjugated to the wavefront in 1 step. In the real conditions, there are many factors interfering with this, such as the hysteresis of the piezoceramics of the mirror, the inaccuracy of measuring the response functions of the wavefront corrector, the inaccuracy of measuring the wavefront to be corrected, the time delay caused by the duration of the calculations and the reaction time of the mirror. In our experience, the system installs the optimal set of voltages for the wavefront correction in 2-3 steps.

The determination of the characteristics of the system that operates in a closed loop is quite a difficult task. It is possible to estimate the AOS control bandwidth by the width of the initial aberration spectrum, which the system can correct. Our AOS effectively corrects for the first 23 Zernike polynomials (excluding tilts) - ##3 – 23. The coefficient of suppression varies from 12 to 3 dB. Since the bandwidth of these aberrations is about 100 Hz, we can assume that the control bandwidth of the closed AOS is at least the same. In fact, our system is able to correct for the higher order wavefront aberrations that change with the frequency higher than 100 Hz but we might face the limitation connected to the ability of our bimorph mirror simply to represent some of these high order aberration – the number of control electrodes in our case was limited to 31.

A graph of the spectral energy for the first 8 Zernike polynomials has been added, from which follows both a diagram of the frequency band occupied by a particular polynomial and the amplitude of the energy for each polynomial.

About Strehl ratio. The measurement (and calculation) of the Strehl coefficient in the strict sense presents certain difficulties, since it is associated with the maximum intensity of the ideal and the real beams. Therefore, based on your suggestion we added the graphs of the changes of the residual error of the wavefront over time to the text of the paper.

A diagram of the internal structure of the FPGA and the time characteristics of the adaptive optical system are now also presented and discussed in this version of the paper.

[1] Oya, S., Bouvier, A., Guyon, O., Watanabe, M., Hayano, Y., Takami, H., Iye, M., Hattori, M., Saito, Y., Itoh, M., Colley, S., Dinkins, M., Eldred, M., Golota, T., "Performance of the deformable mirror for Subaru LGSAO," Proceedings of the SPIE, Volume 6272, pp. 62724S (2006).

Round 2

Reviewer 1 Report

The authors have addressed my previous specific comments, and have made good progress towards addressing my main overall concern which was lack of novelty. 
Bimorph mirrors are not standard in AO systems designed for real time atmospheric correction, but are still used. The Subaru Telescope uses one, for example. This paper should be highlighting the use of FPGAs for the control loop which is the aspect that will be of interest to readers. I would strongly suggest highlighting the use of FPGAs in the title and abstract, as the other aspects of the experiment are largely standard adaptive optics. 

Specific comments:
- The introduction is better, but still lacks context for this experiment. If AO is already being used for real applications, why is this work still needed? What is different?
- Is the bimorph custom made, or commercially available?
- Figure 7, can you provide specific exposure times? 
- Lines 171-172, "Figures 9 and 10" should be "Figures 8 and 9" 
- Figures 8 and 9, are they using the same data? Could this be merged into a single plot?
- Figure 11 implies that for all the modes the DM is able to correct, a bandwidth of ~100Hz should be sufficient, so is this a rigorous enough test for your system running at 2kHz? Also, the caption of this figure doesn't make sense.
- Discussion and conclusion: focus more on the FPGA performance and how well it improves the control loop timing error. A comparison with a CPU based control loop would be very useful if possible.

Author Response

Thank you so much for your comments. They are very useful for us. Regarding the questions, we can answer the following:

1) We tried to put some extra words in the abstract of the paper about the use of FPGA and the advantage we got. As for the title we would like to keep it as it is – in fact FPGA-based systems in general should be interesting for readers, but here we want to pay attention also on the analyses of the behavior of different wavefront aberrations made using a fast adaptive optical system.

2) Done. We have added a paragraph in the introduction emphasizing the use of FPGA.

3) Since bimorph mirrors are designed to work in a specific installation, all mirrors are manufactured in our laboratory and are custom made. And for sure it is possible to order such a mirror according to the individual requirements of the customer (diameter, number of electrodes, coating, etc.)

4) Done. The intensity of the image in the far zone is quite high, so the set of neutral filters were used. Depending on this, the exposure value was selected. For example, for the images of Fig. 7 "Not corrected", the exposure time was 1/11765 s = 85 microseconds, while in the case of correction, the shutter had to be reduced to 1/17544 s = 57 microseconds.

5) Done. We have corrected the wrong picture numbers. Thank you very much for your attention and remark, sorry….

6) Initially, we tried to combine Figures 8 and 9. But since the scale along the vertical axis of these figures differs significantly (by an order of magnitude), the combination leads to the poor visibility of the graph details. In particular, the graphs with subtracted tilts become almost straight lines at the very bottom of the coordinate grid. Therefore, this paper presents two different graphs.

7) When choosing the correction frequency, we were guided by the fact that an ideal discrete system that works with the delay begins to correct for the aberrations at a sampling frequency exceeding the frequency of the input sine signal by 10 times (see attached file Figures 1-3 from “Rukosuev A.L., Kudryashov A.V., Lylova A.N., Samarkin V.V., Sheldakova Yu.V., "Adaptive optical system for real-time wavefront correction", Atmospheric and oceanic optics, 28(4), pp. 381-386, 2015”). Accordingly, we wanted to have some margin for speed, so the sampling frequency was chosen 2000 Hz.

8) Done. We add some FPGA-based AOS advantages to Discussion section

Reviewer 2 Report

The paper has been improved upon since the first version by adding a considerable amount of information on the setup. 

While the proof of concept that turbulence can be corrected in  lab setup using adaptive optics is till not overwhelmingly exciting these days, it is interesting to see what correction loop frequencies can be achieved nowadays, and with what kind of equipment. So I think the paper merits publication now.

I have one remaining question, which the Authors may or may not decide to clarify:

  Fig.6:  Is that for a single lenslet, or averaged across all of them? It would be very interesting to see the closed-loop curve in the same graph, and even to compare the suppression of temporal frequencies.  This could hint at the actual bandwidth of the system. It could then also be related to Figs. 11 and 14.

Author Response

Thank you so much for your comments. They are very useful for us. Regarding the questions, we can answer the following:

1) The power spectral density graph in Fig. 6 was obtained by processing a sample of the X coordinate of the lenslet array focal spot offset for one lens. No averaging was performed, but a comparison was made with similar graphs obtained for several other microlenses. The results were almost identical.

2) Some similar results for our setup we published in SPIE paper [A. L. Rukosuev, V. N. Belousov, A. N. Nikitin, Yu. V. Sheldakova, A. V. Kudryashov, V. A. Bogachev, M. V. Volkov, S. G. Garanin, F. A. Starikov, "Wavefront correction of laser beam distorted by fan heater turbulence using an adaptive optical system with a frequency of 2000 Hz", Proc. SPIE 11678, pp.116780P, 2021. doi: 10.1117/12.2578143]. In this paper, we mainly wanted to analyze the temporal behavior of Zernike polynomials using a fast FPGA-based adaptive optical system in the mode of correction of the laser beam wavefront distorted by air flow
